# An Empirical and Comparative Analysis of Data Valuation with Scalable Algorithms

## Abstract

This paper focuses on valuating training data for supervised learning tasks and studies the Shapley value, a data value notion originated in cooperative game theory. The Shapley value defines a unique value distribution scheme that satisfies a set of appealing properties desired by a data value notion. However, the Shapley value requires exponential complexity to calculate exactly. Existing approximation algorithms, although achieving great improvement over the exact algorithm, relies on retraining models for multiple times, thus remaining limited when applied to larger-scale learning tasks and real-world datasets.

In this work, we develop a simple and efficient heuristic for data valuation based on the Shapley value with complexity independent with the model size. The key idea is to approximate the model via a $K$-nearest neighbor ($K$NN) classifier, which has a locality structure that can lead to efficient Shapley value calculation. We evaluate the utility of the values produced by the $K$NN proxies in various settings, including label noise correction, watermark detection, data summarization, active data acquisition, and domain adaption. Extensive experiments demonstrate that our algorithm achieves at least comparable utility to the values produced by existing algorithms while significant efficiency improvement. Moreover, we theoretically analyze the Shapley value and justify its advantage over the leave-one-out error as a data value measure.

## 1 Introduction

*Data valuation* addresses the question of how to decide the worth of data. Data analysis based on machine learning (ML) has enabled various applications, such as targeted advertisement, autonomous driving, and healthcare, and creates tremendous business values; at the same time, there lacks a principled way to attribute these values to different data sources. Thus, recently, the problem of data valuation has attracted increasing attention in the research community. In this work, we focus on valuing data in relative terms when the data is used for supervised learning.

The Shapley value has been proposed to value data in recent works (Jia et al., 2019b;a; Ghorbani & Zou, 2019). The Shapley value originates from cooperative game theory and is considered a classic way of distributing total gains generated by the coalition of a set of players. One can formulate supervised learning as a cooperative game between different training points and thus apply the Shapley value to data valuation. An important reason for employing the Shapley value is that it *uniquely* possesses a set of appealing properties desired by a data value notion, such as fairness and additivity of values in multiple data uses.

Algorithmically, the Shapley value inspects the marginal contribution of a point to every possible subset of training data and averages the marginal contributions over all subsets. Hence, computing the Shapley value is very expensive and the exact calculation has exponential complexity. The existing works on the Shapley value-based data valuation have been focusing on how to scale up the Shapley value calculation to large training data size. The-state-of-art methods to estimate the Shapley value for general models are based on Monte Carlo approximation (Jia et al., 2019b; Ghorbani & Zou, 2019). However, these methods require to re-train the ML model for a large amount of times; thus, they are only applicable to simple models and small training data size. The first question we ask in the paper is: How can we value data when it comes to large models, such as neural networks, and massive datasets?

In this paper, we propose a simple and efficient heuristic for data valuation based on the Shapley value for any given classification models. Our algorithm obviates the need to retrain the model for multiple times. The complexity of the algorithm scales quasi-linearly with the training data size, linearly with the validation data size, and independent of the model size. The key idea underlying our algorithm is to approximate the model by a $K$-nearest neighbor classifier, whose locality structure can reduce the number of subsets examined for computing the Shapley value, thus enabling tremendous efficiency improvement.

Moreover, the existing works argue the use of the Shapley value mainly based on interpreting its properties (e.g., fairness, additivity) in the data valuation context. However, can we move beyond these known properties and reason about the "performance" of the Shapley value as a data value measure? In this paper, we formalize two performance metrics specific to data values: one focuses on the predictive power of a data value measure, studying whether it is indicative of a training point's contribution to some random set; the other focuses on the ability of a data value to discriminate "good" training points from "bad" ones for privacy-preserving models. We consider leave-one-out error as a simple baseline data value measure and investigate the advantage of the Shapley value in terms of the two above performance metrics.

Finally, we show that our algorithm is a versatile and scalable tool that can be applied to a wide range of tasks: correcting label noise, detecting watermarks used for claiming the ownership of a data source, summarizing a large dataset, guiding the acquisition of new data, and domain adaptation. On small datasets and models where the complexity of the existing algorithms is acceptable, we demonstrate that our approximation can achieve at least comparable performance on these tasks. We also experiment on large datasets and models in which case prior algorithms all fail to value data in a reasonable amount of time and highlight the scablability of our algorithm.

## 2 General Frameworks for Data Valuation

We present two frameworks to interpret the value of each training point for supervised learning and discuss their computational challenges.

We first set up the notations to characterize the main ingredients of a supervised learning problem, including the training and testing data, the learning algorithm and the performance measure. Let $D = \{z_i\}_{i=1}^N$ be the training set, where $z_i$ is a feature-label pair $(x_i, y_i)$, and $D_{\mathrm{val}}$ be the testing data. Let $\mathcal{A}$ be the learning algorithm which maps a training dataset to a model. Let $U$ be a performance measure which takes as input training data, any learning algorithm, and validation data and returns a score. We write $U(S, \mathcal{A}, D_{\mathrm{val}})$ to denote the performance score of the model trained on a subset $S$ of training data using the learning algorithm $\mathcal{A}$ when testing on $D_{\mathrm{val}}$. When the learning algorithm and validation data are self-evident, we will suppress the dependence of $U$ on them and just use $U(S)$ for short. Our goal is to assign a score to each training point $z_i$, denoted by $\nu(z_i, D, \mathcal{A}, D_{\mathrm{val}}, U)$, indicating its *value* for the supervised learning problem specified by $D, \mathcal{A}, D_{\mathrm{val}}, U$. We will often write it as $\nu(z_i)$ or $\nu(z_i, U)$ to simplify notation.

### 2.1 Leave-One-Out Method

One simple way of appraising each training point is to measure its contribution to the rest of the training data:

$$\nu_{\mathrm{loo}}(z_i) = U(D) - U(D \setminus \{z_i\}) \tag{1}$$

This value measure is referred to as the Leave-one-out (LOO) value. The exact evaluation of the $LOO$ values for $N$ training points requires to re-train the model for $N$ times and the associated computational cost is prohibitive for large training datasets and large models. For deep neural networks, Koh & Liang (2017) proposed to estimate the model performance change due to the removal of each training point via influence functions. However, in order to obtain the influence functions, one will need to evaluate the inverse of the Hessian for the loss function. With $N$ training points and $p$ model parameters, it requires $\mathcal{O}(N \times p^2 + p^3)$ operations. Koh & Liang (2017) introduced a method to approximate the influence function with $\mathcal{O}(Np)$ complexity, which is still expensive for large networks. In contrast, our method, which will be discussed in Section 3, has complexity independent of model size, thus preferable for models like deep neural networks.

## 2.2 SHAPLEY VALUE-BASED METHOD

The Shapley value is a classic concept in cooperative game theory to distribute the total gains generated by the coalition of all players. One can think of a supervised learning problem as a cooperative game among training data instances and apply the Shapley value to value the contribution of each training point.

Given a performance measure $U$, the Shapley value for training data $z_i$ is defined as the average marginal contribution of $z_i$ to all possible subsets of $D$ formed by other training points:

$$\nu_{\text{shap}}(z_i) = \frac{1}{N} \sum_{S \subseteq D \setminus \{z_i\}} \frac{1}{\binom{N-1}{|S|}} \left[ U(S \cup \{z_i\}) - U(S) \right] \tag{2}$$

The reason why the Shapley value is appealing for data valuation is that it *uniquely* satisfies the following properties.

  i **Group rationality**: The utility of the machine learning model is completely distributed among all training points, i.e., $U(D) = \sum_{z_i \in D} \nu(z_i)$. This is a natural rationality requirement because any rational group of data contributors would expect to distribute the full yield of their coalition.
  ii **Fairness**: (1) Two data points which have identical contributions to the model utility should have the same value. That is, if for data $z_i$ and $z_j$ and any subset $S \subseteq D \setminus \{z_i, z_j\}$, we have $U(S \cup \{i\}) = U(S \cup \{j\})$, then $\nu(z_i) = \nu(z_j)$. (2) Data points with zero marginal contributions to all subsets of the training set should be given zero value, i.e., $\nu(z_i) = 0$ if $U(S \cup \{z_i\}) = 0$ for all $S \subseteq D \setminus \{z_i\}$.
  iii **Additivity**: When the overall performance measure is the sum of separate performance measures, the overall value of a datum should be the sum of its value under each performance measure $\nu(z_i, U_1) + \nu(z_i, U_2) = \nu(z_i, U_1 + U_2)$ for $z_i \in D$. In machine learning applications, the model performance measure is often evaluated by summing up the individual loss of validation points. We expect the value of a data point in predicting multiple validation points to be the sum of the values in predicting each validation point.

Despite the desirable properties of the Shapley value, calculating the Shapley value is expensive. Evaluating the exact Shapley value involves computing the marginal contribution of each training point to all possible subsets, which is $\mathcal{O}(2^N)$. Such complexity is clearly impractical for valuating a large number of training points. Even worse, for ML tasks, evaluating the utility function *per se* (e.g., testing accuracy) is computationally expensive as it requires to re-train an ML model.

Ghorbani & Zou (2019) introduced two approaches to approximating the Shapley value based on Monte Carlo approximation. The central idea behind these approaches is to treat the Shapley value of a training point as its expected contribution to a random subset and use sample average to approximate the expectation. By the definition of the Shapley value, the random set has size 0 to $N - 1$ with equal probability (corresponding to the $1/N$ factor) and is also equally likely to be any subset of a given size (corresponding to the $\binom{N-1}{|S|}$ factor). In practice, one can implement an equivalent sampler by drawing a random permutation of the training set. Then, the Shapley value can be estimated by computing the marginal contribution of a point to the points preceding it and averaging the marginal contributions across different permutations. However, these Monte Carlo-based approaches cannot circumvent the need to re-train models and therefore are not viable for large models. In our experiments, we found that the approaches in Ghorbani & Zou (2019) can manage data size up to one thousand for simple models such as logistic regression and shallow neural networks, while failing to estimate the Shapley value for larger data sizes and deep nets in a reasonable amount of time. We will evaluate runtime in more details in Section 5.

## 3   SCALABLE DATA VALUATION VIA $K$NN PROXIES

In this section, we explicate our proposed method to achieve efficient data valuation for large training data size and large models like deep nets. The key idea is to approximate the model with a $K$NN, which enjoys efficient algorithms for computing both the LOO and the Shapley value due to its unique locality structure.

### 3.1   $K$NN SHAPLEY VALUE

Given a single validation point $x_{\text{val}}$ with the label $y_{\text{val}}$, the simplest, unweighted version of a $K$NN classifier first finds the top-$K$ training points $(x_{\alpha_1}, \cdots, x_{\alpha_K})$ that are most similar to $x_{\text{val}}$ and outputs

the probability of $x_{\text{val}}$ taking the label $y_{\text{val}}$ as $P[x_{\text{val}} \to y_{\text{val}}] = \frac{1}{K} \sum_{i=1}^{K} \mathbb{1}[y_{\alpha_i} = y_{\text{val}}]$. We assume that the confidence of predicting the right label is used as the performance measure, i.e.,

$$U(S) = \frac{1}{K} \sum_{k=1}^{\min\{K,|S|\}} \mathbb{1}[y_{\alpha_k(S)} = y_{\text{val}}] \tag{3}$$

where $\alpha_k(S)$ represents the index of the training feature that is $k$th closest to $x_{\text{val}}$ among the training examples in $S$. Particularly, $\alpha_k(D)$ is abbreviated to $\alpha_k$. Under this performance measure, the Shapley value can be calculated exactly using the following theorem.

**Theorem 1** (Jia et al. (2019a))**.** *Consider the model performance measure in (3). Then, the Shapley value of each training point can be calculated recursively as follows:*

$$\nu(z_{\alpha_N}) = \frac{\mathbb{1}[y_{\alpha_N} = y_{val}]}{N} \tag{4}$$

$$\nu(z_{\alpha_i}) = \nu(z_{\alpha_{i+1}}) + \frac{\mathbb{1}[y_{\alpha_i} = y_{val}] - \mathbb{1}[y_{\alpha_{i+1}} = y_{val}]}{K} \frac{\min\{K, i\}}{i} \tag{5}$$

Theorem 1 can be readily extended to the case of multiple validation points, wherein the utility function is defined as the average of the utility function with respect to each validation point. By the additivity property, the Shapley value with respect to multiple validation points is the average across the Shapley value with respect to every single validation point. We will call the values obtained from (4) and (5) *the KNN Shapley value* hereinafter. For each validation point, computing the $K$NN Shapley value requires only $\mathcal{O}(N \log N)$ time, which circumvents the exponentially large number of utility evaluations entailed by the Shapley value definition. The intuition for achieveing such exponential improvement is that for $K$NN, the marginal contribution $U(S \cup z_i) - U(S)$ only depends on the relative distance of $z_i$ and $K$-nearest neighbors in $S$ to the validation point. When calculating the Shapley value, instead of considering all $S \subseteq D \setminus \{z_i\}$, we only need to focus on the subsets that result in distinctive $K$-nearest neighbors.

## 3.2 An Efficient Algorithm Enabled by the $K$NN Shapley value

By leveraging the $K$NN Shapley value as a proxy, we propose the following algorithm to value the training data importance for general models that do not enjoy efficient Shapley value calculation methods. For deep nets, the algorithm proceeds as follows: (1) Given the training set, we train the network and obtain the deep features (i.e., the input to the last softmax layer); (2) We train a $K$NN classifier on the deep features and corresponding labels and further calibrate $K$ such that the resulting $K$NN mimics the performance of the original deep net. (3) With a proper choice of $K$ obtained from the last step, we employ Theorem 1 to compute the Shapley value of the deep features. For other models, the algorithm directly computes the $K$NN Shapley value on the raw data as a surrogate for the true Shapley value.

The complexity of the above algorithm is $\mathcal{O}(Nd + N \log N)$ where $d$ is the dimension of deep feature representation. As opposed to Monte Carlo-based methods (e.g., Ghorbani & Zou (2019); Jia et al. (2019b)), the proposed algorithm does not require to retrain models. It is well-suited for approximating values for large models as its complexity is independent of model size.

Note that when applied to deep nets, this algorithm neglects the contribution of data points for feature learning as the feature extractor is fixed therein. In other words, the data values produced by this algorithm attempt to distribute the total yield of a cooperative game between deep features, rather than raw data. However, as we will later show in the experiment, these values can still reflect data usefulness in various applications, while being much more efficient than the existing works.

## 4 Theoretical Comparison Between LOO and the Shapley Value

One may ask how we choose between the LOO method and the Shapley value for valuing training data in machine learning tasks. We have seen that the Shapley value uniquely satisfies several appealing properties. Moreover, prior work (Ghorbani & Zou, 2019) has demonstrated empirical evidence that the Shapley value is more effective than the LOO value for assessing the quality of training data instances. Nevertheless, can we theoretically justify the "valuation performance" of the two value measures?

### 4.1 Predictive Power of the Value Measures

To justify that the data values produced by a valuation technique can reflect the data usefulness in practice, existing valuation techniques are often examined in terms of their performance to work as a pre-processing step to filter out low-quality data, such as mislabeled or noisy data, in a given dataset. Then, one may train a model based only on the remaining "good" data instances or their combination with additional data. Note that both the LOO and the Shapley value only measure the worth of a data point relative to other points in the given dataset. Since it is still uncertain what data will be used in tandem with the point being valued after data valuation is performed, we hope that the value measures of a point are indicative of the expected performance boost when combining the point with a random set of data points.

In particular, we consider two points that have different values under a given value measure and study whether the expected model performance improvements due to the addition of these two points will have the same order as the estimated values. With the same order, we can confidently select the higher-value point in favor of another when performing ML tasks. We formalize this desirable property in the following definition.

**Definition 1.** *We say a value measure $\nu$ to be order-preserving at a pair of training points $z_i, z_j$ that have different values if*

$$\left(\nu(z_i, U) - \nu(z_j, U)\right) \times \mathbb{E}\left[U(T \cup \{z_i\}) - U(T \cup \{z_j\})\right] > 0 \tag{6}$$

*where $T$ is an arbitrary random set drawn from some distribution.*

For general model performance measures $U$, it is difficult to analyze the order-preservingness of the corresponding value measures. However, for $K$NN, we can precisely characterize this property for both the LOO and the Shapley value. The formula for the $K$NN Shapley value is given in Theorem 1 and we present the expression for the $K$NN LOO value in the following lemma.

**Lemma 1** ($K$NN LOO Value). *Consider the model performance measure in (3). Then, the $K$NN LOO value of each training point can be calculated by $\nu_{loo}(z_{\alpha_i}) = \frac{1}{K}\left(\mathbb{1}[y_{\alpha_i} = y_{val}] - \mathbb{1}[y_{\alpha_{K+1}} = y_{val}]\right)$ if $i \leq K$ and $0$ otherwise.*

Now, we are ready to state the theorem that exhibits the order-preservingness of the $K$NN LOO value and the $K$NN Shapley value.

**Theorem 2.** *For any given $D = \{z_1, \ldots, z_N\}$, where $z_i = (x_i, y_i)$, and any given validation point $z_{val} = (x_{val}, y_{val})$, assume that $z_1, \ldots, z_N$ are sorted according to their similarity to $x_{val}$. Let $d(\cdot, \cdot)$ be the feature distance metric according to which $D$ is sorted. Suppose that $P_{(X,Y) \in \mathcal{D}}(d(X, x_{val}) \geq d(x_i, x_{val})) > \delta$ for all $i = 1, \ldots, N$ and some $\delta > 0$. Then, $\nu_{shap\text{-}knn}$ is order-preserving for all pairs of points in $I$; $\nu_{LOO\text{-}knn}$ is order-preserving only for $(z_i, z_j)$ such that $\max i, j \leq K$.*

Due to the space limit, we will omit all proofs to the appendix. The assumption that $P_{(X,Y) \in \mathcal{D}}(d(X, x_{\text{val}}) \geq d(x_i, x_{\text{val}})) > \delta$ in Theorem 2 intuitively means that it is possible to sample points that are further away from $x_{\text{val}}$ than the points in $D$. This assumption can easily hold for reasonable data distributions in continuous space.

Theorem 2 indicates that the $K$NN Shapley value has more predictive power than the $K$NN LOO value—the $K$NN Shapley value can predict the relative utility of any two points in $D$, while the $K$NN LOO value is only able to correctly predict the relative utility of the $K$-nearest neighbors of $x_{\text{val}}$. In Theorem 2, the relative data utility of two points is measured in terms of the model performance difference when using them in combination with a random dataset.

Theorem 2 can be generalized to the setting of multiple validation points using the additivity property. Specifically, for any two training points, the $K$NN Shapley value with respect to multiple validation points is order-preserving when the order remains the same on each validation point, while the $K$NN LOO value with respect to multiple validation points is order-preserving when the two points are within the $K$-nearest neighbors of all validation points and the order remains the same on each validation point. We can see that similar to the single-validation-point setting, the condition for the $K$NN LOO value with respect to multiple validation points to be order-preserving is more stringent than that for the KNN Shapley value.

Moreover, we would like to highlight that the definition of order-preservingness is proposed as a property for data value measures; nevertheless, it can also be regarded as a property of a data value measure estimator. A data value estimator will be order-preserving if the estimation error is much

smaller than the minimum gap between the data value measures of any two points in the training set. Since the estimation error of a consistent estimator can be made arbitrarily small as long with enough samples, a consistent estimator (if exists) for an order-preserving data value measure is also order-preserving when the sample size is large. An example for such estimator is the sample average of the marginal contribution of a point to the ones preceding it in multiple random permutations.

## 4.2 Usability for Differentially Private Algorithms

Since the datasets used for machine learning tasks often contain sensitive information (e.g., medical records), it has been increasingly prevalent to develop privacy-preserving learning algorithms. Hence, it is also interesting to study how to value data when the learning algorithm preserves some notion of privacy. Differential privacy (DP) has emerged as a strong privacy guarantee for algorithms on aggregate datasets. The idea of DP is to carefully randomize the algorithm so that the output does not depend too much on any individuals' data.

**Definition 2** (Differential privacy). $\mathcal{A} : \mathcal{D}^N \to \mathcal{H}$ is $(\epsilon, \delta)$-differentially private if for all $R \subseteq \mathcal{H}$ and for all $D, D' \in \mathcal{D}^N$ such that $D$ and $D'$ differ only in one data instance: $P[\mathcal{A}(D) \in R] \leq e^\epsilon P[\mathcal{A}(D') \in R] + \delta$.

By definition, differential private learning algorithms will hide the influence of one training point on the model performance. Thus, intuitively, it will be more difficult to differentiate "good" data from "bad" ones for differentially private models. We will show that the Shapley value could have more discriminative power than the LOO value when the learning algorithms satisfy DP.

The following theorem states that for differentially private learning algorithms, the values of training data are gradually indistinguishable from dummy points set as the training size grows larger using both the LOO and the Shapley value measures; nonetheless, the value differences vanish faster for the LOO value than the Shapley value.

**Theorem 3.** *For a learning algorithm $\mathcal{A}(\cdot)$ that achieves $(\epsilon(N), \delta(N))$-DP when training on $N$ data points. Let the performance measure be $U(S) = -\frac{1}{M} \sum_{i=1}^{M} \mathbb{E}_{h \sim \mathcal{A}(S)} l(h, z_{val,i})$ for $S \subseteq D$. Let $\epsilon'(N) = e^{c\epsilon(N)} - 1 + ce^{c\epsilon(N)}\delta(N)$. Then, it holds that*

$$\max_{z_i \in D} \nu_{loo}(z_i) \leq \epsilon'(N-1) \qquad \max_{z_i \in D} \nu_{shap}(z_i) \leq \frac{1}{N-1} \sum_{i=1}^{N-1} \epsilon'(i) \qquad (7)$$

For typical differentially private learning algorithms, such as adding random noise to stochastic gradient descent, the privacy guarantees will be weaker if we reduce the size of training set (e.g., see Theorem 1 in Abadi et al. (2016)). In other words, $\epsilon(n)$ and $\delta(n)$ are monotonically decreasing functions of $n$, and so is $\epsilon'(n)$. Therefore, it holds that $\epsilon'(N) < \frac{1}{N} \sum_{i=1}^{N} \epsilon'(i)$. The implications of Theorem 3 are three-fold. Firstly, the fact that the maximum difference between all training points' values and the dummy point's value is directly upper bounded by $\epsilon'$ signifies that stronger privacy guarantees will naturally lead to the increased difficulty to distinguish between data values. Secondly, the monotonic dependence of $\epsilon'$ on $N$ indicates that both the LOO and the Shapley value are converging to zero when the training size is very large. Thirdly, by comparing the upper bound for the LOO and the Shapley value, we can see that the convergence rate of the Shapley value is slower and therefore it could have a better chance to differentiate between "good" data from the bad than the LOO value.

Note that our results are extendable to general *stable* learning algorithms, which are insensitive to the removal of an arbitrary point in the training dataset (Bousquet & Elisseeff, 2002). Stable learning algorithms are appealing as they enjoy provable generalization error bounds. Indeed, differentially private algorithms are subsumed by the class of stable algorithms (Wang et al., 2015). A broad variety of other learning algorithms are also stable, including all learning algorithms with Tikhonov regularization. We will leave the corresponding theorem for stable algorithms to Appendix C.

## 5 Experiments

In this section, we first compare the runtime of our algorithm with the existing works on various dataset. Then, we compare the usefulness of the data values produced by different algorithms based on various applications, including mislabeled data detection, watermark removal, data summarization, active data acquisition, and domain adaptation. We will leave the detailed experimental setting, such as model architecture and hyperparamters of training processes, to the appendix.

## 5.1 BASELINES

We will compare our algorithm, termed KNN-Shapley, with the following baselines.

**Truncated Monte Carlo Shapley (TMC-Shapley).** This is a Monte Carlo-based approximation algorithm proposed by Ghorbani & Zou (2019). Monte Carlo-based methods regard the Shapley value as the expectation of a training instance's marginal contribution to a random set and then use the sample mean to approximate it. Evaluating the marginal contribution to a different set requires to retrain the model, which bottlenecks the efficiency of Monte Carlo-based methods. TMC-Shapley combined the Monte Carlo method with a heuristic that ignores the random sets of large size since the contribution of a data point to those sets will be small.

**Gradient Shapley (G-Shapley).** This is another Monte Carlo-based method proposed by Ghorbani & Zou (2019) and employs a different heuristic to accelerate the algorithm. G-Shapley approximates the model performance change due to the addition of one training point by taking a gradient descent step using that point and calculating the performance difference. This method is applicable only to the models trained with gradient methods; hence, the method will be included as a baseline in our experimental results when the underlying models are trained using gradient methods.

**Leave-One-Out.** We use Leave-one-out to refer to the algorithm that calculates the exact model performance due to the removal of a training point. Evaluating the exact leave-one-out error requires to re-train the model on the corresponding reduced dataset for every training point, thus also impractical for large models.

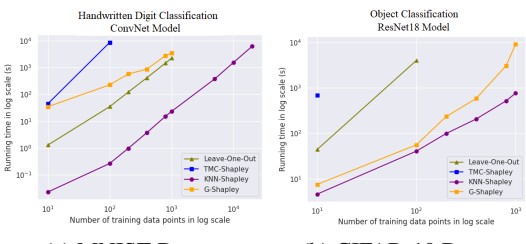

(a) MNIST Dataset    (b) CIFAR-10 Dataset

Figure 1: Comparison of running time of different data valuation algorithms. (a) is implemented on a machine with 2.6GHz CPU and 32GB memory and (b) is implement on a machine with 1.80GHz and 32GB memory

**KNN-LOO** Leave-one-out is nevertheless efficient for $K$NN as shown in Theorem 1. To use the KNN-LOO for valuing data, we first approximate the target model with a $K$NN and compute the corresponding $K$NN-LOO value. If the model is a deep net, we compute the value on the deep feature representation; otherwise, we compute the value on the raw data.

**Random** The random baseline does not differentiate between different data's value and just randomly selects data points from training set to perform a given task.

## 5.2 RUNTIME COMPARISON

We compare their runtime on different datasets and models and exhibit the result for a logistic regression trained on MNIST and a larger model ResNet-18 on CIFAR-10 in Figure 1. We can see that $K$NN Shapley outperforms the rest of baselines by several orders of magnitude for large training data size and large model size.

## 5.3 APPLICATIONS

Most of the following applications are discussed in recent work on data valuation (Ghorbani & Zou, 2019). In this paper, we hope to understand: *Can a simple, scalable heuristic to approximate the Shapley value using a KNN surrogate outperforms these, often more computationally expensive, previous approaches on the same set of applications?* As a result, our goal is not to outperform state-of-the-art methods for each application; instead, we hope to put our work in the context of current efforts in understanding the relationships between different data valuation techniques and their performance on these tasks.

**Detecting Noisy Labels** Labels in the real world are often noisy due to automatic labeling, non-expert labeling, or label corruption by data poisoning adversaries. Even if a human expert can identify incorrectly labeled examples, it is impossible to manually verify all labels for large training data. We show that the notion of data value can help human experts prioritize the verification process, allowing them to review only the examples that are most likely to be contaminated. The key idea is to rank the data points according to their data values and prioritize the points with the lowest values. Following (Ghorbani & Zou, 2019), we perform experiments in the three settings: a Naive Bayes model trained on a spam classification dataset, a logistic regression model trained on Inception-

V3 features of a flowe classification dataset, and a three-layer convolutional network trained on fashion-MNIST dataset. The noise flipping ratio is 20%, 10%, and 10%, respectively. The detection performance of different data value measures is illustrated in Figure 2. We can see that the $K$NN Shapley value outperforms other baselines for all the three settings. Also, the Shapley value-based values, including ours, TMC-Shapley, and G-Shapley, are more effective than the LOO-based values.

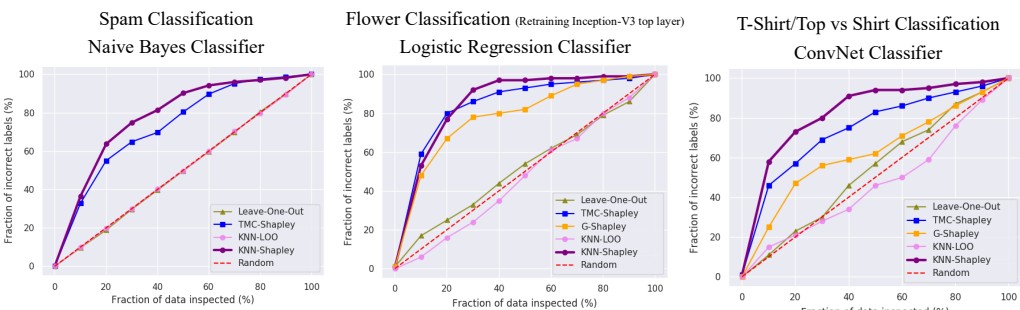

Figure 2: **Detecting noisy labels.** We examine the label for the training points with lowest values and plot the fraction of mislabeled data detected changes with the fraction of training data checked.

**Watermark Removal**    Deep neural networks have achieved tremendous success in various fields but training these models from scratch could be computationally expensive and require a lot of training data. One may contribute a dataset and outsource the computation to a different party. How can the dataset contributor claim the ownership of the data source of a trained model? A prevalent way of addressing this question is to embed watermarks into the DNNs. There are two classes of watermarking techniques in the existing work, i.e., pattern-based techniques and instance-based techniques. Specifically, pattern-based techniques inject a set of samples blended with the same pattern and labeled with a specific class into the training set; the data contributor can later verify the data source of the trained model by checking the output of the model for an input with the pattern. Instance-based techniques, by contrast, inject individual training samples labeled with a specific class as watermarks and the verification can be done by inputting the same samples into the trained model. Some examples of the watermarks generated by the pattern-based and instance-based techniques are illustrated in Figure 7 in the appendix. In this experiment, we will demonstrate that based on the data values, the model trainer is always possible to remove the watermarks. The idea is that the watermarks should have low data values by nature, since they contribute little to predict the normal validation data. Note that this experiment constitutes a new type of attack, which might of independent interest itself.

For the pattern-based watermark removal experiment, we consider three settings: two convolutional networks trained on 1000 images from fashion MNIST and 10000 images from MNIST, resepctively, and a ResNet18 model trained on 1000 images from a face recognition dataset, Pubfig-83. The watermark ratio is 10% for all three settings. The details about watermark patterns are provided in the appendix. Since for the last two settings, either due to large data size or model size, TMC-Shapley, G-Shapley, and Leave-one-out all fail to produce value estimates in 3 hours, we compare our algorithm only with the rest of baselines. The results are shown in Figure 3. We can see that $K$NN-Shapley achieves similar performance to TMC-Shapley when the time complexity of TMC-Shapley is acceptable and outperforms all other baselines.

For the instance-based watermark removal experiment, we consider the following settings: a logistic regression model trained on 10000 images from MNIST, a convolution network trained on 3000 images from CIFAR-10, and ResNet18 trained on 3000 images from SVHN. The watermark ratio is 10%, 3%, and 3%, respectively. The results of our experiment are shown in Figure 4. For this experiment, we found that both watermarks and benign data tend to have low values on some validation points; therefore, watermarks and benign data are not quite separable in terms of the average value across the validation set. We propose to compute the max value across the validation set for each training point, which we call max-KNN-Shapley, and remove the points with lowest max-KNN-Shapley values. The intuition is that out-of-distribution samples are inessential to predict all normal validation points and thus the maximum of their Shapley values with respect to different validation points should be low. The results show that the max-KNN-Shapley is more effective measure to detect instance-based watermarks than all other baselines.

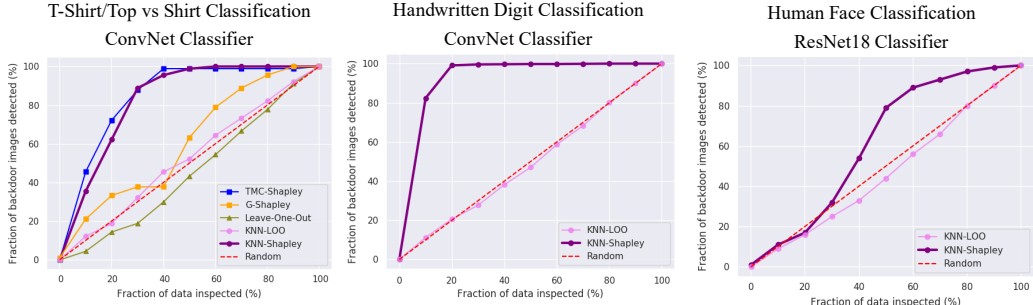

Figure 3: **Pattern-based watermark removal.** We examine the label for the training points with lowest values and plot the fraction of watermarks detected with the fraction of training data checked. For the last two settings, TMC-Shapley, G-Shapley, and Leave-one-out are omitted because they cannot finish in 3 hours.

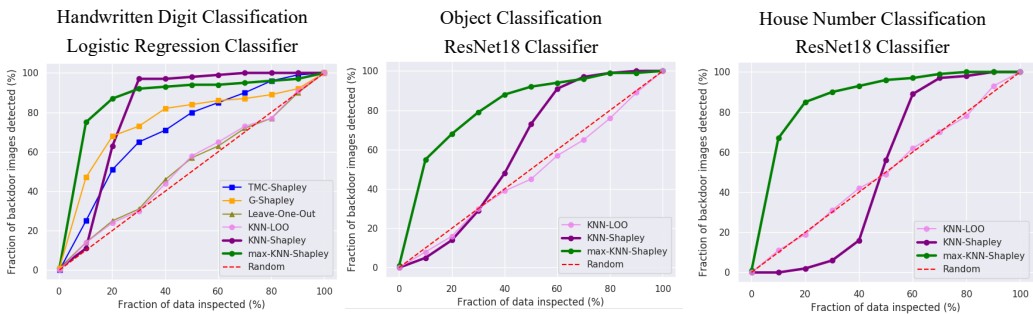

Figure 4: **Instance-based watermark removal.** We examine the label for the training points with lowest values and plot the faction of watermarks detected with the fraction of training data checked. The max-KNN-Shapley in the plot calculates the maximum of the $K$NN Shapley values across all validation points.

**Data Summarization** Data summarization aims to find a small subset that well represents a massive dataset. The utility of the succinct summaries should be comparable to that of the whole dataset. This is a natural application of data values, since we can just summarize the dataset by removing low-value points. We consider two settings for this experiment: a single hidden layer neural network trained on UCI Adult Census dataset and a ResNet-18 trained on Tiny ImageNet. For the first setting, we randomly pick 1000 individuals as a training set, where half of them have income exceeding $50,000$ per year. We use another a balanced dataset of size 500 to calculate the values of training data and 1000 data points to evaluate the model performance. As Figure 5 (a) shows, our method remains a high performance even reducing $50\%$ of the whole training set. The data selected by the Shapley value-based data values are more helpful to boost model performance than the LOO-based value measures. TMC-Shapley and G-Shapley can achieve slightly better performance than $K$NN-Shapley. In the second setting, we use 95000 points as the training set, 5000 points to calculate the values, and 10000 points as the held-out set. The result in Figure 5 (b) shows that $K$NN-Shapley is able to maintain model performance even after removing 40% of the whole training set. However, TMC-Shapley, G-Shapley, and LOO cannot finish in 24 hours and hence are omitted from the figure.

**Active Data Acquisition** Annotated data is often hard and expensive to obtain, particularly for specialized domains where only experts can provide reliable labels. Active data acquisition aims to ease the data collection process by automatically deciding which instances an annotator should label to train a model as efficiently and effectively as possible. We assume that we start with a small training set and compute the data values. Then, we train a random forest to predict the value for new data based on their features and select the points with highest values. For this experiment, we consider two setups. For the first setup, we synthesize a dataset with disparate data qualities by add noise to partial MNIST. For the second, we use Tiny ImageNet, which has realistic variation of data quality, for evaluation. In the first setup, we choose 100 images from MNIST and add Gaussian white noise into half of them. We use another 100 images to calculate the training data values and a held-out dataset of size 1000 to evaluate the performance. In the second setup, we separate the training set into two parts with 5000 training points and 95000 new points. We calculate values of

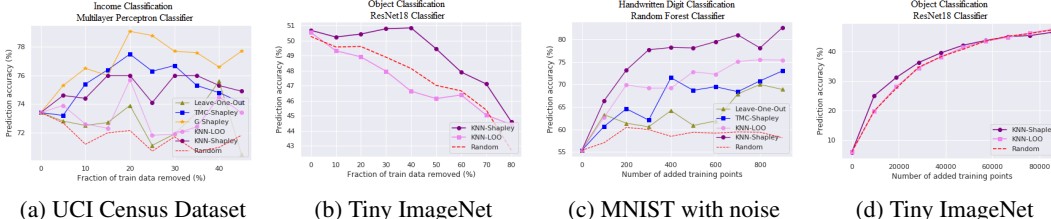

| (a) UCI Census Dataset | (b) Tiny ImageNet | (c) MNIST with noise | (d) Tiny ImageNet |

Figure 5: (a-b) **Data summarization**. We remove low-value points from the ground training set and evaluate the accuracy of the model trained with remaining data. (c-d) **Active data acquisition**. We compute the data values for a small starting set, train a Random forest to predict the values of new data, and then add the new points with highest values into the starting set. We plot the model accuracy change when combining the starting set with more and more new data added according to different value measures.

2500 data points in the training set based on the other 2500 points. Both Figure 5 (c) and (d) show that new data selected based on $K$NN-Shapley value can improve model accuracy faster than the rest of baselines.

**Domain Adaptation** Machine learning models are know to have limited capablity of generalizing learned knowledge to new datasets or environments. In practice, there is a need to transfer the model from a source domain where sufficient training data is available to a target domain where few labeled data are available. Domain adaptation aims to better leverage the dataset from one domain for the prediction tasks in another domain. We will show that data values will be useful for this task. Specifically, we can compute the values of data in the source domain with respect to a held-out set from the target domain. In this way, the data values will reflect how useful different training points are for the task in the target domain. We then train the model based only on positive-value points in the source domain and evaluate the model performance in the target one. We perform experiments on three datasets, namely, MNIST, USPS, and SVHN. For the transfer between USPS and MNIST, we use the same experiment setting as (Ghorbani & Zou, 2019). We firstly train a multinomial logistic regression classifier. We randomly sample 1000 images from the source domain as training set, calculate the values for the training data based on 1000 instances from the target domain, and evaluate the performance of the model on another 1000 target domain data. The results are summarized in Table 1, which shows that $K$NN-Shapley performs better than TMC-Shapley. For the transfer between SVHN and MNIST, we pick 2000 training data from SVHN, train a ResNet-18 model (He et al., 2016), and evaluate the performance on the whole test set of MNIST. $K$NN-Shapley is able to implement on the data of this scale efficiently while TMC-Shapley algorithm cannot finish in 48 hours.

Table 1: **Domain adapation**. We calculate the value of data in source domain based on a validation set from the target domain, and then pick the points with positive values to train the model for performing prediction tasks in the target domain.

| Method | MNIST → USPS | USPS → MNIST | SVHN → MNIST |
|---|---|---|---|
| $K$NN-Shapley | $31.7\% \rightarrow \mathbf{48.40\%}$ | $23.35\% \rightarrow \mathbf{30.25\%}$ | $9.65\% \rightarrow \mathbf{20.25\%}$ |
| TMC-Shapley | $31.7\% \rightarrow 44.90\%$ | $23.35\% \rightarrow 29.55\%$ | - |
| $K$NN-LOO | $31.7\% \rightarrow 39.40\%$ | $23.35\% \rightarrow 24.52\%$ | $9.65\% \rightarrow 11.70\%$ |
| LOO | $31.7\% \rightarrow 29.40\%$ | $23.35\% \rightarrow 23.53\%$ | - |

## 6 CONCLUSION

In this paper, we propose an efficient algorithm to approximate the Shapley value based on $K$NN proxies, which for the first time, enables the data valuation for large-scale dataset and large model size. We demonstrate the utility of the approximate Shapley values produced by our algorithm on a variety of applications, from noisy label detection, watermark removal, data summarization, active data acquisition, to domain adaption. We also compare with the existing Shapley value approximation algorithms, and show that our values can achieve comparable performance while the computation is much more efficient and scalable. We characterize the advantage of the Shapley value over the LOO value from a theoretical perspective and show that it is preferable in terms of predictive power and discriminative power under differentially private learning algorithms.

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

# A  PROOF OF THEOREM 2

**Theorem 2.** *For any given $D = \{z_1, \ldots, z_N\}$, where $z_i = (x_i, y_i)$, and any given validation point $z_{val} = (x_{val}, y_{val})$, assume that $z_1, \ldots, z_N$ are sorted according to their similarity to $x_{val}$. Let $d(\cdot, \cdot)$ be the feature distance metric according to which $D$ is sorted. Suppose that $P_{(X,Y) \in \mathcal{D}}(d(X, x_{val}) \geq d(x_i, x_{val})) > \delta$ for all $i = 1, \ldots, N$ and some $\delta > 0$. Then, $\nu_{shap\text{-}knn}$ is order-preserving for all pairs of points in $I$; $\nu_{LOO\text{-}knn}$ is order-preserving only for $(z_i, z_j)$ such that $\max i, j \leq K$.*

*Proof.* The proof relies on dissecting the term $\mathbb{E}[U(T \cup \{z_i\}) - U(T \cup \{z_j\})]$ and $\nu(z_i) - \nu(z_j)$ ($\nu = \nu_{shap\text{-}knn}, \nu_{LOO\text{-}knn}$) in the definition of order-preserving property.

Consider any two points $z_i, z_{i+l} \in D$. We start by analyzing $\mathbb{E}[U(T \cup \{z_i\}) - U(T \cup \{z_{i+l}\})]$. Let the $k$th nearest neighbor of $x_{val}$ in $T$ be denoted by $T_{(k)} = (x_{(k)}, y_{(k)})$. Moreover, we will use $T_{(k)} \leq_d z_i$ to indicate that $x_{(k)}$ is closer to the validation point than $x_i$, i.e., $d(x_{(k)}, x_{val}) \leq d(x_i, x_{val})$. We first analyze the expectation of the above utility difference by considering the following cases:

(1) $T_{(K)} \leq_d z_i$. In this case, adding $z_i$ or $z_{i+l}$ into $T$ will not change the $K$-nearest neighbors to $z_{\text{val}}$ and therefore $U(T \cup \{z_i\}) = U(T \cup \{z_{i+l}\}) = U(T)$. Hence, $U(T \cup \{z_i\}) - U(T \cup \{z_{i+l}\}) = 0$.

(2) $z_i <_d T_{(K)} \leq_d z_{i+l}$. In this case, including the point i into T can expel the Kth nearest neighbor from the original set of K nearest neighbors while including the point $i + 1$ will not change the $K$ nearest neighbors. In other words, $U(T \cup \{z_i\}) - U(T) = \frac{\mathbb{1}[y_i = y_{\text{val}}] - \mathbb{1}[y_{(K)} = y_{\text{val}}]}{K}$ and $U(T \cup \{z_{i+l}\}) - U(T) = 0$. Hence, $U(T \cup \{z_i\}) - U(T \cup \{z_{i+l}\}) = \frac{\mathbb{1}[y_i = y_{\text{val}}] - \mathbb{1}[y_{(K)} = y_{\text{val}}]}{K}$.

(3) $T_{(K)} >_d z_{i+l}$. In this case, including the point $i$ or $i + 1$ will both change the original $K$ nearest neighbors in $T$ by excluding the $K$th nearest neighbor. Thus, $U(T \cup \{z_i\}) - U(T) = \frac{\mathbb{1}[y_i = y_{\text{val}}] - \mathbb{1}[y_{(K)} = y_{\text{val}}]}{K}$ and $U(T \cup \{z_{i+l}\}) - U(T) = \frac{\mathbb{1}[y_{i+l} = y_{\text{val}}] - \mathbb{1}[y_{(K)} = y_{\text{val}}]}{K}$. It follows that $U(T \cup \{z_i\}) - U(T \cup \{z_{i+l}\}) = \frac{\mathbb{1}[y_i = y_{\text{val}}] - \mathbb{1}[y_{i+l} = y_{\text{val}}]}{K}$.

Combining the three cases discussed above, we have

$$\mathbb{E}[U(T \cup \{z_i\}) - U(T \cup \{z_{i+l}\})] \tag{8}$$

$$= P(T_{(K)} \leq_d z_i) \times 0 + P(z_i <_d T_{(K)} \leq_d z_{i+l}) \frac{\mathbb{1}[y_i = y_{\text{val}}] - \mathbb{1}[y_{(K)} = y_{\text{val}}]}{K}$$

$$+ P(T_{(K)} >_d z_{i+l}) \frac{\mathbb{1}[y_i = y_{\text{val}}] - \mathbb{1}[y_{i+l} = y_{\text{val}}]}{K} \tag{9}$$

$$= P(z_i <_d T_{(K)} \leq_d z_{i+l}) \frac{\mathbb{1}[y_i = y_{\text{val}}] - \mathbb{1}[y_{(K)} = y_{\text{val}}]}{K}$$

$$+ P(T_{(K)} >_d z_{i+l}) \frac{\mathbb{1}[y_i = y_{\text{val}}] - \mathbb{1}[y_{i+l} = y_{\text{val}}]}{K} \tag{10}$$

Note that removing the first term in (10) cannot change the sign of the sum in (10). Hence, when analyzing the sign of (10), we only need to focus on the second term:

$$P(T_{(K)} >_d z_{i+1}) \frac{\mathbb{1}[y_i = y_{\text{val}}] - \mathbb{1}[y_{i+l} = y_{\text{val}}]}{K} \tag{11}$$

Since $P(T_{(K)} >_d z_{i+1}) = \sum_{k=N-K+1}^{N} P(Z >_d z_{i+1})^k$, the sign of (11) will be determined by the sign of $\mathbb{1}[y_i = y_{\text{val}}] - \mathbb{1}[y_{i+l} = y_{\text{val}}]$. Hence, we get

$$\left(\mathbb{E}[U(T \cup \{z_i\}) - U(T \cup \{z_{i+1}\})]\right) \times \left(\mathbb{1}[y_i = y_{\text{val}}] - \mathbb{1}[y_{i+l} = y_{\text{val}}]\right) > 0 \tag{12}$$

Now, we switch to the analysis of the value difference. By Theorem 1, it holds for the $K$NN Shapley value that

$$\nu_{\text{shap-knn}}(z_i) - \nu_{\text{shap-knn}}(z_{i+l}) \tag{13}$$

$$= \sum_{j=i}^{i+l-1} \frac{\min\{K, j\}}{jK} \left(\mathbb{1}[y_j = y_{\text{val}}] - \mathbb{1}[y_{j+1} = y_{\text{val}}]\right) \tag{14}$$

$$= \frac{\min\{K, i\}}{iK} \mathbb{1}[y_i = y_{\text{val}}] + \sum_{j=i}^{i+l-2} \left(\frac{\min\{K, j+1\}}{(j+1)K} - \frac{\min\{K, j\}}{jK}\right) \mathbb{1}[y_{j+1} = y_{\text{val}}]$$

$$- \frac{\min\{K, i+l-1\}}{(i+l-1)K} \mathbb{1}[y_{i+l} = y_{\text{val}}] \tag{15}$$

Note that $\frac{\min\{K, j+1\}}{(j+1)K} - \frac{\min\{K, j\}}{jK} < 0$ for all $j = i, \ldots, i + l - 2$. Thus, if $\mathbb{1}[y_i = y_{\text{val}}] = 1$ and $\mathbb{1}[y_{i+l} = y_{\text{val}}] = 0$, the minimum of (15) is achieved when $\mathbb{1}[y_{j+1} = y_{\text{val}}] = 1$ for all $j = i, \ldots, i + l - 2$ and the minimum value is $\frac{\min\{K, i+l-1\}}{(i+l-1)K}$, which is greater than zero. On the other hand, if $\mathbb{1}[y_i = y_{\text{val}}] = 0$ and $\mathbb{1}[y_{i+l} = y_{\text{val}}] = 1$, then the maximum of (15) is achieved when $\mathbb{1}[y_{j+1} = y_{\text{val}}] = 0$ for all $j = i, \ldots, i + l - 2$ and the maximum value is $-\frac{\min\{K, i+l-1\}}{(i+l-1)K}$, which is less than zero.

Summarizing the above analysis, we get that $\nu_{\text{shap-knn}}(z_i) - \nu_{\text{shap-knn}}(z_{i+l})$ has the same sign as $\mathbb{1}[y_i = y_{\text{val}}] - \mathbb{1}[y_{i+l} = y_{\text{val}}]$. By (12), it follows that $\nu_{\text{shap-knn}}(z_i) - \nu_{\text{shap-knn}}(z_{i+l})$ also shares the same sign as $\mathbb{E}[U(T \cup \{z_i\}) - U(T \cup \{z_{i+1}\})]$.

To analyze the sign of the $K$NN LOO value difference, we first write out the expression for the $K$NN LOO value difference:

$$\nu_{\text{loo-knn}}(z_i) - \nu_{\text{loo-knn}}(z_{i+l}) = \begin{cases} \frac{1}{K}(\mathbb{1}[y_i = y_{\text{val}}] - \mathbb{1}[y_{i+l} = y_{\text{val}}]) & \text{if } i + l \leq K \\ \frac{1}{K}(\mathbb{1}[y_i = y_{\text{val}}] - \mathbb{1}[y_{K+1} = y_{\text{val}}]) & \text{if } i \leq K < i + l \\ 0 & \text{if } i > K \end{cases} \qquad (16)$$

Therefore, $\nu_{\text{loo-knn}}(z_i) - \nu_{\text{loo-knn}}(z_{i+l})$ has the same sign as $\mathbb{1}[y_i = y_{\text{val}}] - \mathbb{1}[y_{i+l} = y_{\text{val}}]$ and $\mathbb{E}[U(T \cup \{z_i\}) - U(T \cup \{z_{i+1}\})]$ only when $i + l \leq K$.

$\square$

## B  PROOF OF THEOREM 3

We will need the following lemmas on group differential privacy for the proof of Theorem 3.

**Lemma 2.** *If $\mathcal{A}$ is $(\epsilon, \delta)$-differentially private with respect to one change in the database, then $\mathcal{A}$ is $(c\epsilon, ce^{c\epsilon}\delta)$-differentially private with respect to $c$ changes in the database.*

**Lemma 3** (Jia et al. (2019b)). *For any $z_i, z_j \in D$, the difference in Shapley values between $z_i$ and $z_j$ is*

$$\nu_{shap}(z_i) - \nu_{shap}(z_j) = \frac{1}{N-1} \sum_{T \subseteq D \setminus \{z_i, z_j\}} \frac{U(T \cup \{z_i\}) - U(T \cup \{z_j\})}{\binom{N-2}{|T|}} \qquad (17)$$

**Theorem 3.** *For a learning algorithm $\mathcal{A}(\cdot)$ that achieves $(\epsilon(N), \delta(N))$-DP when training on $N$ data points. Let the performance measure be $U(S) = -\frac{1}{M}\sum_{i=1}^{M} \mathbb{E}_{h \sim \mathcal{A}(S)} l(h, z_{val,i})$ for $S \subseteq D$. Let $\epsilon'(N) = e^{c\epsilon(N)} - 1 + ce^{c\epsilon(N)}\delta(N)$. Then, it holds that*

$$\max_{z_i \in D} \nu_{loo}(z_i) \leq \epsilon'(N-1) \qquad\qquad \max_{z_i \in D} \nu_{shap}(z_i) \leq \frac{1}{N-1} \sum_{i=1}^{N-1} \epsilon'(i) \qquad (7)$$

*Proof.* Let $S'$ be the set with one element in $S$ replaced by a different value. Let the probability density/mass defined by $\mathcal{A}(S')$ and $\mathcal{A}(S)$ be $p(h)$ and $p'(h)$, respectively. Using Lemma 2, for any $z_{\text{val}}$ we have

$$\mathbb{E}_{h \sim \mathcal{A}(S)} l(h, z_{\text{val}}) = \int_0^1 P_{h \sim \mathcal{A}(S)}[l(h, z_{\text{val}}) > t] dt \qquad (18)$$

$$\leq \int_0^1 (e^{c\epsilon} P_{h \sim \mathcal{A}(S')}[l(h, z_{\text{val}}) > t] + ce^{c\epsilon}\delta) dt \qquad (19)$$

$$= e^{c\epsilon} \mathbb{E}_{h \sim \mathcal{A}(S')}[l(h, z_{\text{val}})] + ce^{c\epsilon}\delta \qquad (20)$$

It follows that

$$\mathbb{E}_{h \sim \mathcal{A}(S)} l(h, z_{\text{val}}) - \mathbb{E}_{h \sim \mathcal{A}(S')}[l(h, z_{\text{val}})] \leq (e^{c\epsilon} - 1)\mathbb{E}_{h \sim \mathcal{A}(S')}[l(h, z_{\text{val}})] + ce^{c\epsilon}\delta \qquad (21)$$

$$\leq e^{c\epsilon} - 1 + ce^{c\epsilon}\delta \qquad (22)$$

By symmetry, it also holds that

$$\mathbb{E}_{h \sim \mathcal{A}(S')} l(h, z_{\text{val}}) - \mathbb{E}_{h \sim \mathcal{A}(S)}[l(h, z_{\text{val}})] \leq (e^{c\epsilon} - 1)\mathbb{E}_{h \sim \mathcal{A}(S)}[l(h, z_{\text{val}})] + ce^{c\epsilon}\delta \qquad (23)$$

$$\leq e^{c\epsilon} - 1 + ce^{c\epsilon}\delta \qquad (24)$$

Thus, we have the following bound:

$$|\mathbb{E}_{h \sim \mathcal{A}(S)} l(h, z_{\text{val}}) - \mathbb{E}_{h \sim \mathcal{A}(S')}[l(h, z_{\text{val}})]| \leq e^{c\epsilon} - 1 + ce^{c\epsilon}\delta \qquad (25)$$

Denoting $\epsilon' = e^{c\epsilon} - 1 + ce^{c\epsilon}\delta$. For the performance measure that evaluate the loss averaged across multiple validation points $U(S) = -\frac{1}{M}\sum_{i=1}^{M}\mathbb{E}_{h\sim\mathcal{A}(S)}l(h, z_{\mathrm{val},i})$, we have

$$|U(S) - U(S')| \leq \epsilon' \tag{26}$$

Making the dependence on the training set size explicit, we can re-write the above equation as

$$\max_{z_i, z_j \in D, T \subseteq D\setminus\{z_i, z_j\}} |U(T \cup z_i) - U(T \cup z_j)| \leq \epsilon'(|T| + 1) \tag{27}$$

By Lemma 3, we have for all $z_i, z_j \in D$,

$$\nu_{\mathrm{shap}}(z_i) - \nu_{\mathrm{shap}}(z_j) \leq \frac{1}{N-1}\sum_{k=0}^{N-2}\sum_{T\subseteq D\setminus\{z_i,z_j\},|T|=k}\frac{\epsilon'(k+1)}{\binom{N-2}{k}} \tag{28}$$

$$= \frac{1}{N-1}\sum_{k=0}^{N-2}\epsilon'(k+1) \tag{29}$$

$$= \frac{1}{N-1}\sum_{k=1}^{N-1}\epsilon'(k) \tag{30}$$

As for the LOO value, we have

$$\nu_{loo}(z_i) - \nu_{loo}(z_j) = U(D\setminus\{z_j\}) - U(D\setminus\{z_i\}) \tag{31}$$

$$\leq \epsilon'(N-1) \tag{32}$$

$\square$

## C    COMPARING THE LOO AND THE SHAPLEY VALUE FOR STABLE LEARNING ALGORITHMS

An algorithm $G$ has uniform stability $\gamma$ with respect to the loss function $l$ if $\|l(G(S), \cdot) - l(G(S^{\setminus i}), \cdot)\|_\infty \leq \gamma$ for all $i \in \{1, \cdots, |S|\}$, where $S$ denotes the training set and $S^{\setminus i}$ denotes the one by removing $i$th element of $S$.

**Theorem 4.** *For a learning algorithm $\mathcal{A}(\cdot)$ with uniform stability $\beta = \frac{C_{stab}}{|S|}$, where $|S|$ is the size of the training set and $C_{stab}$ is some constant. Let the performance measure be $U(S) = -\frac{1}{M}\sum_{i=1}^{M}l(A(S), z_{val,i})$. Then,*

$$\max_{z_i \in D}\nu_{loo}(z_i) - \nu_{loo}(z^*) \leq \frac{C_{stab}}{N-1} \tag{33}$$

*and*

$$\max_{z_i \in D}\nu_{shap}(z_i) - \nu_{shap}(z^*) \leq \frac{C_{stab}(1 + \log(N-1))}{N-1} \tag{34}$$

*Proof.* By the definition of uniform stability, it holds that

$$\max_{z, z_j \in D, T\subseteq D\setminus\{z_i, z_j\}} |U(T \cup \{z_i\}) - U(T \cup \{z_j\})| \leq \frac{C_{\mathrm{stab}}}{|T| + 1} \tag{35}$$

Using Lemma 3, we have we have for all $z_i, z_j \in D$,

$$\nu_{\mathrm{shap}}(z_i) - \nu_{\mathrm{shap}}(z_j) \tag{36}$$

$$\leq \frac{1}{N-1}\sum_{k=0}^{N-2}\sum_{T\subseteq D\setminus\{z_i,z_j\},|T|=k}\frac{C_{\mathrm{stab}}}{\binom{N-2}{k}(k+1)} \tag{37}$$

$$= \frac{1}{N-1}\sum_{k=0}^{N-2}\frac{C_{\mathrm{stab}}}{k+1} \tag{38}$$

Recall the bound on the harmonic sequences

$$\sum_{k=1}^{N} \frac{1}{k} \leq 1 + \log(N)$$

which gives us

$$\nu_{\text{shap}}(z_i) - \nu_{\text{shap}}(z_j) \leq \frac{C_{\text{stab}}(1 + \log(N-1))}{N-1}$$

As for the LOO value, we have

$$\nu_{loo}(z_i) - \nu_{loo}(z_j) = U(D \setminus \{z_j\}) - U(D \setminus \{z_i\}) \leq \frac{C_{\text{stab}}}{N-1} \tag{39}$$

$\square$

# D  ADDITIONAL EXPERIMENTS

## D.1  REMOVING DATA POINTS OF HIGH VALUE

We remove training points from most valuable to least valuable and evaluate the accuracy of the model trained with remaining data. The experimental setting, including datasets and models, is the same as that for the data summarization experiment. Figure 6 compares the power of different data value measures to detect high-utility data. We can see that removing high-value points based on KNN-Shapley, G-Shapley, and TMC-Shapley can all significantly reduce the model performance. It indicates that these three heuristics are effective in detecting most valuable training points. On UCI Census, TMC-Shapley achieves the best performance and $K$NN-Shapley performs similarly to G-Shapley. On Tiny ImageNet, both TMC-Shapley and G-Shapley cannot finish in 24 hours and are therefore omitted from comparison. Compared with the random baseline, $K$NN-Shapley can lead to a much faster performance drop when removing high-value points.

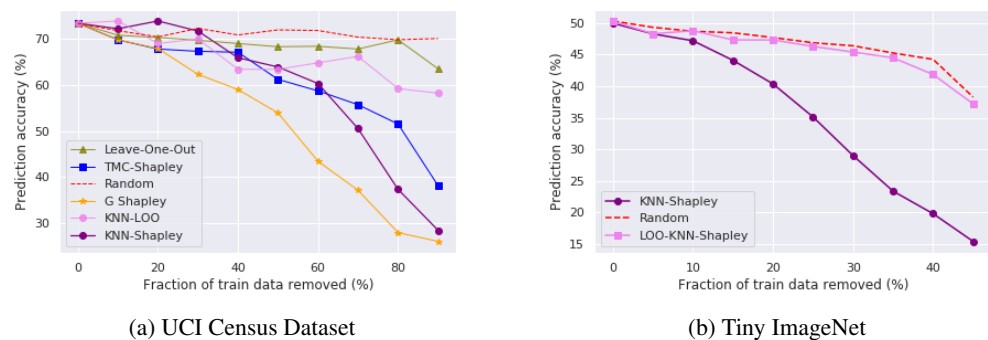

(a) UCI Census Dataset                    (b) Tiny ImageNet

Figure 6: (a-b) We remove high-value points from the ground training set and evaluate the accuracy of the model trained with remaining data.

## D.2  RANK CORRELATION WITH GROUND TRUTH SHAPLEY VALUE

We perform experiments to compare the ground truth Shapley value of raw data and the value estimates produced by different heuristics. The ground truth Shapley value is computed using the group testing algorithm in (Jia et al., 2019b), which can approximate the Shapley value with provable error bounds. We use a fully-connected neural network with three hidden layers as the target model. Following the setting in (Jia et al., 2019b), we construct a size-1000 training set using MNIST, which contains both benign and adversarial examples, as well as a size-100 validation set

with pure adversarial examples. The adversarial examples are generated by the Fast Gradient Sign Method (Goodfellow et al., 2014). This construction is meant to simulate data with different levels of usefulness. In the above setting, the adversarial examples in the training set should be more valuable than the benign data because they can improve the prediction on adversarial examples. Note that the $K$NN-Shapley computes the Shapley value of deep features extracted from the penultimate layer.

The rank correlation of $K$NN-Shapley and G-Shapley with the ground truth Shapley value is 0.08 and 0.024 with p-value 0.0046 and 0.4466, respectively. It shows that both heuristics may not be able to preserve the exact rank of the ground truth Shapley value. Since TMC-Shapley cannot finish in a week for this model and data size, we omit it from comparison. We further apply some local smoothing to the values and check whether these heuristics can produce large values for data groups with large Shapley values. Specifically, we compute 1 to 100 percentiles of the Shapley values, find the group of data points within each percentile interval, and compute the average Shapley value as well as the average heuristic values for each group. The rank correlation of the average $K$NN-Shapley and the average G-Shapley with the average ground truth Shapley value for these data groups are 0.22 and -0.002 with p-value 0.0293, 0.9843, respectively. We can see that although ignoring the data contribution for feature learning, $K$NN-Shapley can better preserve the rank of the Shapley value in a macroscopic level than G-Shapley.

## E  EXPERIMENT DETAILS

### E.1  PATTERN-BASED WATERMARK REMOVAL

We adopted two types of patterns: one is to change the pixel values at the corner of an image Chen et al. (2018a), another is to blend a specific word (like "TEST") in an image, as shown in Figure 7a. The first pattern is used in the experiments on fashion MNIST and MNIST, which contain single channel images. The second pattern is used in the experiment on Pubfig-83, which contains multi-channel images.

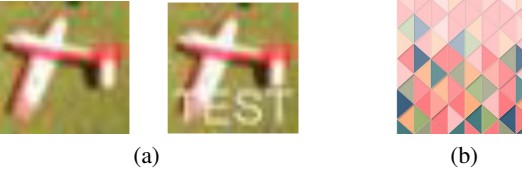

|  | (a) | (b) |

Figure 7: (a) Examples of pattern-based watermarks. Specifically, after an image is blended with the "TEST" pattern, it is classified as the target label, e.g, an "automobile" on CIFAR-10. (b) Example of instance-based watermarks, which are typically chosen as some out-of-distribution data with specific assigned labels.

Table 2 shows the prediction accuracy of the watermarked model on both benign and watermark instances. The results indicate that the amount of watermarks we added can help successfully claim the ownership of the data source.

| Model Accuracy | Handwritten Digit (Logistic Regression) | Object (ResNet18) | House Number (ResNet18) |
|---|---|---|---|
| Watermarked Model on Benign Data | 0.997778 | 0.980667 | 1 |
| Watermarked Model on Watermark Data | 0.98 | 1 | 1 |

Table 2: Model accuracy for pattern-based watermark removal experiment.

### E.2  INSTANCE-BASED WATERMARK REMOVAL

We used the same watermarks as Adi et al. (2018), which contains a set of abstract images with specific assigned labels. The example of a trigger image is shown in Figure 7b.

### E.3 Data Summarization

For the experiment on UCI Adult Census dataset, we train the same multilayer perceptron mode as Chen et al. (2018b). The network architecture is displayed in Table 3.

| Model for Adult Census Dataset |
| --- |
| FC(6) + Sigmoid |
| FC(100) + Sigmoid |

Table 3: Model for evaluating the data meaning in Data Summarization

For the experiment on Tiny ImageNet, we fine-tune the pretrained ResNet18 from He et al. (2016). We train the ResNet18 with 15 epochs and learning rate 0.001 and the model achieves an accuracy of 77.95% on the training set. Then, we extract the deep features of the training set and calculate their Shapley values. When evaluating the model performance on the summarized dataset, we re-train the ResNet18 with 30 epochs and learning rate 0.01.

### E.4 Domain Adaptation

As to the domain adaptation between MNIST and USPS, we train a multinomial logistic regression classifier. For transfer from SVHN to MNIST, we train a ResNet18 model using 15 epochs and learning rate 0.001 on SVHN, since multinomial logistic regression is too simple to perform well in this setting.

