# OpenReview forum: "An Empirical and Comparative Analysis of Data Valuation with Scalable Algorithms"
_ICLR.cc/2020/Conference — Reject_

### Official Review · AnonReviewer1 · 2019-10-23
**Official Blind Review #1**

**Rating:** 1

**Review:**

Given that computing the Shapely value for data valuation is very expensive and existing approximate methods are not scalable either, the authors introduce a new approach based on K-NN approximation of the model to scale it specifically for DNN. The authors propose to use the final features produced in the last feature extractor layer of DNN as features for KNN and choose K such that the performance of KNN is closest to the performance of DNN. My main problem with this approach is that the authors still need to do this for any trained DNN in order to compute a good value for Equation (2). If the claim here is that the features extractor layers of deep neural network does not change by changing the training set (which is a huge claim), then why one should use K-NN. We can simply use the feature extractor part of DNN (almost all the trainable parameters except the last layer) once and then fix it and only learn the soft-max layer parameter for different subsets. Overall, I believe even though this paper aims to address an important problem, the approach is taken is not well-justified and lacks value. Below  are some other minor problems:

The introduction/title of the paper claims this is a general approach for any model but the authors' focus is only on DNN. This should be corrected.

Inconsistent notation:
Beginning of Section 2. The training and test set is first denoted by D and D_{test} and then later by S and S_{test}.
Equation (2): the authors are using U in a different forms that the ones introduced earlier in Section 2. I recommend the authors only introduce one notation for U and stick with it throughout the paper.

Writing problems:
Section 2: “algorithm algorithm” → algorithm (repeated word)
 Section 2: “For each training data z_i, our goal is to assign a score to each training point, denoted by … ” →Our goal is to assign a score to each training data z_i denoted by …

Note that the assumption in machine learning is that you do not have access to the test set and it is something you won’t see until you deployed your method. I assume the authors meant validation set.

Constant C is introduced in Equation (2) but it is not well justified.


**Experience Assessment:**

I have read many papers in this area.

**Review Assessment: Checking Correctness Of Derivations And Theory:**

N/A

**Review Assessment: Checking Correctness Of Experiments:**

I did not assess the experiments.

**Review Assessment: Thoroughness In Paper Reading:**

I read the paper thoroughly.

---

> ### Author Response · Authors · 2019-11-11
> **Response to Reviewer #1 (Part 1)**
>
> We thank the reviewer for the comments.
>
> Q: The authors propose to use the final features produced in the last feature extractor layer of DNN as features for KNN and choose K such that the performance of KNN is closest to the performance of DNN. My main problem with this approach is that the authors still need to do this for any trained DNN in order to compute a good value for Equation (2). If the claim here is that the features extractor layers of deep neural network does not change by changing the training set (which is a huge claim), then why one should use K-NN. We can simply use the feature extractor part of DNN (almost all the trainable parameters except the last layer) once and then fix it and only learn the soft-max layer parameter for different subsets. Overall, I believe even though this paper aims to address an important problem, the approach is taken is not well-justified and lacks value.
>
>
> A: We agree that fixing the feature extraction layer is a compromise; however, re-training feature extractor for a large number of times is known to be intractable, so our goal is to understand whether our simple heuristic can help tackle the computational challenges and produce a useful data value measure. We believe our experiments have already shown that even fixing the feature extractor, the resulting data values are still useful while being much more efficient than other existing methods. As far as we know, KNN is the only classifier that enjoys scalable algorithms for computing the Shapley value. This motivates us to investigate whether it can be used as a proxy for computing the Shapley value for other classifiers. Moreover, during the rebuttal period, we performed a new experiment to examine the rank correlation between the KNN Shapley of deep features with the ground truth Shapley value of raw data for a neural network. Our experiments show that the values produced by this simple heuristic preserves the rank of ground truth shapley value better than other much more time-consuming heuristics in the existing works. Please see more details in Appendix D.2). Hence, our work takes a step towards practical and scalable data valuation and we believe our work is well-justified and has values to the community.
>
> Moreover, we agree with the reviewer that we can devise another heuristic, which is to estimate the Shapley values of deep features for the last soft-max layer (equivalent to a logistic regression classifier). However, we are not aware of any algorithms for logistic regression that can scale to data size considered in our paper. The state-of-the-art way to estimate the Shapley value for logistic regression is via Monte-Carlo approximation and it requires to re-train logistic regression for a large number of times and the amount of re-training needed grows with the number of training points. Therefore, even with the feature extractor fixed, it is still computational expensive to estimate the Shapley value for the last soft-max layer when the training size is large. That’s why we use KNN instead.
>
> Indeed, we have considered another heuristic which can potentially counts the contribution of data points to feature learning. The heuristic is to extract the feature representations of a given training point at different layers, compute the KNN Shapley value for the feature representation at each layer, and then average the layer-wise KNN Shapley values. However, experimentally, we find that this heuristic achieves comparable performance to the current method which only computes the KNN Shapley value for the feature representation at the second to the last layer, while being more computationally expensive. Therefore, in our paper, we just fix the feature extractor and compute the KNN Shapley value on the deep features extracted by the second to the last layer.
>
> Q: The introduction/title of the paper claims this is a general approach for any model but the authors' focus is only on DNN. This should be corrected.
>
> A: We apologize for not making the experiment details as clear as we intended. Actually, in our experiments, we also examine the models like Naive Bayes and Logistic Regression, which do not enjoy efficient Shapley value calculation methods. Therefore, we use the KNN Shapley value as a surrogate and our experiments show that in general, this heuristic is simple but effective. We clarified in Section 3.2 that for non-DNNs, we directly compute the Shapley value on the raw data as a surrogate for the true Shapley value. We also highlighted the models that we used in the experiments (including both DNNs and non-DNNs) in the title of each figure.

---

> > ### Author Response · Authors · 2019-11-15
> > **Response to Reviewer #1 (Part 2)**
> >
> > Q: Note that the assumption in machine learning is that you do not have access to the test set and it is something you won’t see until you deployed your method. I assume the authors meant validation set.
> >
> > A: Thanks for pointing it out. Yes, we meant the validation set and we have corrected this throughout the paper.
> >
> > Q: Constant C is introduced in Equation (2) but it is not well justified.
> >
> > A: Thank you for pointing it out. Indeed, C=1 in the Shapley value definition. Since we only care about the relative value, we introduced a constant before the Shapley value definition. We can see from the comments that such constant causes unnecessary confusion, so we present the classical Shapley value definition with C=1 in the revised version. Please see Section 2.2. Thanks for the suggestion!
> >
> > We have also polished the writing of the paper and fixed the inconsistent notations.

---

### Official Review · AnonReviewer3 · 2019-10-23
**Official Blind Review #3**

**Rating:** 1

**Review:**

In short, the paper reports improved results on a few applications of Shapley Value of data points using an introduced approximation method that is orders of magnitude faster compared to existing methods.

I vote for rejection of this paper mainly because of two reasons. First, the contributions are not enough for this venue. The paper uses an already existing method (Jia et al. (2019a)) with the only difference being that they use it on top of learned features and therefore the main contribution seems to be the discussions in Sec4. Secondly, the paper makes technically "false" claims (as will be dicussed below).

The positive aspects of the work are as follows: First, the elephant in the room for data valuation methods, which is assessing how good or bad a data point is, is against privacy and this work addresses this question for the first time in the (very small) literature. Secondly, the experimental results are very comprehensive and make a very good case for usefulness of the introduced algorithm. Thirdly, new useful terms for the emerging community of data valuation are introduced through the clear and well-written definitions of the paper.

The paper mentions that the previously introduced KNN-Shapley method applied to the learned features of a deep neural network could be used as an approximation of data points' Shapley values. This is false. All data points contribute value to the feature learning part and the approximation simply ignores this crucial fact. The whole point of using Shapley values a measure of data value is its properties which are not satisfied for the "collaborative game of ML model training" by this approximation; the approximation can be heavily biased due to the fact that ignores the contributions to the feature learning (let's not forget what made deep network's desirable in the first place is their feature extracting power). One cannot use the training data to learn the feature extractor and then ignore the contributions by definition of the Shapley value being the average contribution to a random subset of data points (which means the rest of the data points are removed from the game). Or, one can do such a thing but the method cannot be called an approximation for the true Shapley Value of data points. The G-Shapley heuristic mentioned in the paper from previous work also seems to suffer from the same drawback as it is not playing the same collaborative game of training the ML model (unless one assumes that simply taking one step of the gradient for every data points would be a good approximation for a complete training!)

The experimental results are comprehensive and convincing.  The main issue is that the work discusses these experiments as if the goal of computing data value is performing such tasks (for each of which there exist simpler methods not related to data valuation). " The valuation methods often serve as a preprocessing step to filter out low-quality data,
such as mislabeled or noisy data, in a given dataset" is not a correct statement. The valuation methods serve as valuation methods which the introduced method, although providing " a valuation method", is not providing an unbiased estimate of the equitable Shapley value valuation method. For many of the provided tasks in the experiments section, previous works (Ghorbani & Zou, Jia et al 2019 b) report the same experiments as further inspections into the Shapley Value for data and not as goals of computing these computationally expensive values.

All in all, although the paper's experimental and theoretical results are useful and interesting as the use case of "a valuation method", but is technically incorrect as calling it an approximation for Data Shapley values makes it not publishable. My score is subject to drastic change if a major rework is done to make this point clear.

A few questions and suggestions:

* One of the most striking results from the Data-Shapley works were removing points from most valuable to least valuable and looking at the accuracy drop speed. It would be very necessary and also very convincing if the introduced method is good at detecting very positive points as well as very negative points.
* An interesting empirical experiment would be to look at the Rank Correlation between the introduced approximation and other unbiased Shapley Value approximations. If the correlation is high, it means that empirically the data points contribute equally to feature learning and their value can actually be approximated just by looking at the accuracy on extracted features.
* Is any unbiased estimator of true Shapley Values is "order-preserving" by definition?
* In Sec 2 it would be more useful for the general audience to include the third Shapley property too.
* For almost all of the cases of comparison where previous methods are present for comparison, there seems to be no meaningful advantage. This makes interpreting Figures like 4b, 4c, 5b and most importantly Fig 6b. It would be necessary to add other benchmarks that are not Shapley based. For instance, for data summarization, there has been a line of work the methods of which could be used as a measure of comparison.

**Experience Assessment:**

I have published in this field for several years.

**Review Assessment: Checking Correctness Of Derivations And Theory:**

I carefully checked the derivations and theory.

**Review Assessment: Checking Correctness Of Experiments:**

I carefully checked the experiments.

**Review Assessment: Thoroughness In Paper Reading:**

I read the paper thoroughly.

---

> ### Author Response · Authors · 2019-11-11
> **Response to Reviewer #3 (Part 1)**
>
> We would like to thank the reviewer for the insightful reviews.
>
> Q: The proposed algorithm cannot be called “an approximation of data points' Shapley values” due to the fact that we fix the feature extractor and do not take into account the contribution of data to the feature learning.
>
> A: Thanks for the interesting question! We agree with the reviewer that calling our proposed method as well as other existing work like TMC and G-Shapley in (Ghorbani & Zou) an “approximation of data points’ Shapley values” is a bit misleading, since we ignore the contribution of data points to feature learning. Therefore, we have revised Section 3.2 to highlight that when applied to deep nets, the data value produced by our algorithm attempts to distribute the total yield of a cooperative game between *deep features*, rather than raw data. Moreover, as we show in the experiment, although we do not explicitly count the contribution of feature learning,  these values can reflect data value in various applications, while being much more efficient than the existing works.
>
> We have also considered another heuristic which can potentially take into account the contribution of data points to feature learning. The heuristic is to extract the feature representations of a given training point at different layers of a neural network, compute the KNN Shapley value for the feature representation at each layer,  and then average all layer-wise KNN Shapley values. However, experimentally, we find that this heuristic achieves comparable performance to the current method which only computes the KNN Shapley value for the feature representation at the second to the last layer, while being more computationally expensive.
>
> Q:  The main issue is that the work discusses these experiments as if the goal of computing data value is performing such tasks (for each of which there exist simpler methods not related to data valuation).
>
> A: Thanks for the question! This work is inspired by the observation that most, if not all, recent work on data valuation [1] all provide experimental studies on these applications and tasks (even though there definitely exist simpler methods). In this paper, we hope to understand Can a simple scalable heuristic to calculate Shapley value using a KNN surrogate outperforms these, often more computationally expensive, previous approaches on the same set of tasks?
> As a result, our goal was not to outperform state-of-the-art methods for each task; Instead, we hope to put our work in the context of current efforts in understanding the relationships between different notions of data value and the performance on these tasks. We clarified this point at the beginning of Section 5.3.
>
> Q: “The valuation methods often serve as a preprocessing step to filter out low-quality data,
> such as mislabeled or noisy data, in a given dataset" is not a correct statement.
>
> A: Thanks for pointing it out, and we agree this sentence is confusing. The sentence “The valuation methods often serve as a preprocessing step to filter out low-quality data, such as mislabeled or noisy data, in a given dataset" appears in our theory section 4.2. By this sentence, we really mean that existing works tend to use the experiments, including mislabeled or noisy data identification, to demonstrate that the Shapley value can distinguish data quality and reflect data value in practice. We would like to give some theoretical justification for this empirical observation. We have revised the sentence to eliminate the confusion. Please see Section 4.1.

---

> > ### Author Response · Authors · 2019-11-11
> > **Response to Reviewer #3 (part 2)**
> >
> > Q: One of the most striking results from the Data-Shapley works were removing points from most valuable to least valuable and looking at the accuracy drop speed. It would be very necessary and also very convincing if the introduced method is good at detecting very positive points as well as very negative points.
> >
> > A: Thanks for the interesting question. We have added corresponding results and discussion in Appendix D.1.
> >
> > Q: An interesting empirical experiment would be to look at the Rank Correlation between the introduced approximation and other unbiased Shapley Value approximations. If the correlation is high, it means that empirically the data points contribute equally to feature learning and their value can actually be approximated just by looking at the accuracy on extracted features.
> >
> > A: Thanks for bringing up this interesting experiment. We have added corresponding results to Appendix D.2. In short, we find that although ignoring the data contribution for feature learning, $K$NN-Shapley can better preserve the rank of the Shapley value in a macroscopic level than G-Shapley. Since TMC-Shapley cannot finish in a week for this model and data size, we omit it from comparison.
> >
> > Q: Is any unbiased estimator of true Shapley Values is "order-preserving" by definition?
> >
> > A: Thank you for the interesting question! We would like to highlight that the definition of order-preservingness was originally proposed as a property for data value measures; nevertheless, it can also be regarded as a property of a data value measure estimator. A data value estimator will be order-preserving when the estimation error is much smaller than the minimum gap between the data value measures of any two points in the training set. Since the estimation error of a consistent estimator can be made arbitrarily small as long with enough samples, a consistent estimator (if exists) for an order-preserving data value measure is also order-preserving when the sample size is large. An example for such estimator is the sample average of the marginal contribution of a point to the ones preceding it in multiple random permutations. The estimator in this example is also unbiased. We have added the discussion into the end of Section 4.1.
> >
> > Q: In Sec 2 it would be more useful for the general audience to include the third Shapley property too.
> >
> > A: Thank you for the valuable suggestions, and we have added the third property into the paper in the revised version and please see Section 2.2.
> >
> > Q: For almost all of the cases of comparison where previous methods are present for comparison, there seems to be no meaningful advantage. This makes interpreting Figures like 4b, 4c, 5b and most importantly Fig 6b. It would be necessary to add other benchmarks that are not Shapley based. For instance, for data summarization, there has been a line of work the methods of which could be used as a measure of comparison.
> >
> > A: Thanks for pointing it out. Our main goal is to compare a simple heuristic for computing data value with the existing, often more computationally expensive heuristics. Therefore, we use the same set of tasks considered in the existing papers (Ghorbani & Zou). We do not aim to outperform state-of-the-art methods for each task; Instead, we hope to put our work in the context of current efforts in understanding the relationships between different notions of data value and the performance on these tasks. We will make this clear in the revised version.
> >
> > [1] https://arxiv.org/pdf/1902.10275.pdf

---

> > > ### Comment · AnonReviewer3 · 2019-11-13
> > > **Rebuttal**
> > >
> > > "we will rename our method as deep-feature-KNN-Shapley based on the suggestion. "
> > > As mentioned before, the use of the term "heuristic" for the given algorithm is technically false. Methods like TMC-Shapley (although heuristics), are seeking to approximate the actual "Shapley value" of the data: the Shapley value of the collaborative game among data points as players. The given algorithm is missing the feature learning aspect and therefore cannot be referred to as a heuristic for approximating "Shapley" values of data points. Any use of the concept "Shapley values" for the given heuristic would be misleading as the heuristic is not considering the collaborative game of "supervised learning".
> > >
> > > "Thanks for the interesting question. The accuracy decrease speed of removing points from most valuable to least valuable using our deep-feature-KNN-Shapley heuristic is similar to TMC-Shapley but slightly worse than G-Shapley (See this anonymous link for the result on UCI census dataset https://ibb.co/cQ9NkJX). The result on Tiny ImageNet is still running and we will add them into the revised version. "
> > > These results should be added to the main text as they have become one of the gold-standards in the literature; "valuable points are actually valuable"
> > >
> > > "We are currently performing an experiment which compares the Shapley value estimates of a neural network using the permutation sampling method in [1], which gives provable error guarantees, with the KNN-Shapley value computed on the feature extracted by the first layer. We will update the result later into the revised version. "
> > > These results could at least provide empirical evidence that the heuristic is approximating a value which behaves similar to that of the Shapley value.

---

> > > > ### Author Response · Authors · 2019-11-14
> > > > **Response to Reviewer #3's additional comments**
> > > >
> > > > Thanks a lot for your prompt reply and helpful comments.
> > > >
> > > > Q: As mentioned before, the use of the term "heuristic" for the given algorithm is technically false. Methods like TMC-Shapley (although heuristics), are seeking to approximate the actual "Shapley value" of the data: the Shapley value of the collaborative game among data points as players. The given algorithm is missing the feature learning aspect and therefore cannot be referred to as a heuristic for approximating "Shapley" values of data points. Any use of the concept "Shapley values" for the given heuristic would be misleading as the heuristic is not considering the collaborative game of "supervised learning".
> > > >
> > > > A: Thanks for clarifying your previous comment. We agree with the reviewer that using the term “heuristic” may still be confusing as it overlooks the fact that, when applied to deep neural networks our algorithm fixes the feature extractor for different coalitions of training points. To eliminate such confusion, we will clarify in the revised version that: (1) For deep neural networks, our algorithm provides a heuristic to calculate *the Shapley value of deep features of training data*; we empirically show that the KNN Shapley value of deep features is correlated with that of corresponding raw data. We apply this heuristic to various tasks and demonstrate that the values produced by our algorithm can reflect data value in practice while being much more computationally efficient than the existing works. (2) For other models, since we directly compute the KNN Shapley value on the raw data space, our algorithm provides a heuristic to compute *the Shapley value for raw data*. Overall, our algorithm takes a step towards valuing data for large-scale data and models, a setting that is challenging for all prior works to produce meaningful values in a reasonable amount of time.
> > > >
> > > > Based on the above discussion, we plan to differentiate between raw-data-KNN-Shapley and deep-feature-KNN-Shapley whenever applicable. We are willing to take suggestions from you to decide the best name for our algorithm, we will appreciate it if you have further feedbacks.
> > > >
> > > >
> > > > Q: These results should be added to the main text as they have become one of the gold-standards in the literature; "valuable points are actually valuable"
> > > >
> > > > A: We will add the result into the revised version. Also, as a side note, we have also obtained the result for the tiny ImageNet (this is a scale where G-Shapley and TMC-Shapley don’t finish in 24 hours and are hence omitted from the figure) and attached it in this anonymous link https://ibb.co/WzPS41L
> > > >
> > > > Q: These results could at least provide empirical evidence that the heuristic is approximating a value which behaves similar to that of the Shapley value.
> > > >
> > > > A: We have completed the experiment for comparing the ground truth Shapley value of raw data and the KNN Shapley value of deep features. The ground truth Shapley value is computed using the group testing algorithm in [1], which can approximate the Shapley value with provable error bounds. We used a fully-connected neural network with three hidden layers as the target model. The rank correlation between deep-feature-KNN-Shapley and ground truth Shapley value is 0.08 with p-value 0.0046. It shows that the deep-feature-KNN-Shapley may not be able to preserve the exact rank of the ground truth Shapley value. We further applied some local smoothing to the two values and see whether data groups with large Shapley value also has large deep-feature-KNN-Shapley value. We computed 1-100 percentiles of Shapley values, found the group of data points within each percentile interval (say, between 1st and 2nd percentile), and computed the average Shapley value as well as the average deep-feature-KNN-Shapley value for each group. The rank correlation between average deep-feature-KNN-Shapley and average ground truth Shapley value for these data groups is 0.22 with p-value 0.0293. We can see that deep-feature-KNN-Shapley can preserve the rank of the Shapley value to some extent in a macroscopic level.
> > > >
> > > > We are computing the results for TMC and G-Shapley. Because their speed is slow, the code is still running now. We will compare their rank correlation coefficient with ours in the revised version later.
> > > >
> > > > [1] https://arxiv.org/abs/1902.10275

---

> > > > > ### Author Response · Authors · 2019-11-15
> > > > > **Updated the very first two responses**
> > > > >
> > > > > We updated the very first two responses to incorporate the actual changes we made to the paper for each review comment.
> > > > >
> > > > > We want to thank the reviewer for the helpful comments, which greatly help us improve the manuscript.

---

### Official Review · AnonReviewer2 · 2019-10-23
**Official Blind Review #2**

**Rating:** 3

**Review:**

In this paper, the authors have developed an algorithm to estimate Shapley value with complexity independent of the model size, based on the KNN classifier. Although the paper is interesting in general, and the experiment results are strong, I still feel that the current version of the paper has not quite met the (very high) standard of ICLR, for the following reasons:

1) The authors need to better motivate the advantages of using Shapley value as a data valuation metric. It is not completely clear to me why Shapley value is a good data valuation metric, compared with other options. The authors argue that it is both fair and decomposable (linear in U). However, based on Section 2.2, it is only fair under two extreme cases (identical points and zero marginal contributions). Also, it seems that a lot of other metrics will also satisfy the decomposability condition. Please explain!

2) Section 3 and Section 4.1 focus on U defined in equation (3), in which the testing set is a singleton. It seems to be a major limitation of the paper and it is not clear to me whether or not it is easy to generalize the results in these two sections to the general testing set with multiple points. Please explain!

3) In Definition 3, the definition for the dummy point. This definition requires that U(S \union {z_i}) = U(S) for any S \subseteq D, and in particular it should hold for S=\emptyset. Does U({z_i}) = U(\emptyset) make sense in most practical problems?

A typo: in Definition 2, n and N should be the same.



**Experience Assessment:**

I have read many papers in this area.

**Review Assessment: Checking Correctness Of Derivations And Theory:**

I assessed the sensibility of the derivations and theory.

**Review Assessment: Checking Correctness Of Experiments:**

I assessed the sensibility of the experiments.

**Review Assessment: Thoroughness In Paper Reading:**

I read the paper at least twice and used my best judgement in assessing the paper.

---

> ### Author Response · Authors · 2019-11-11
> **Response to Reviewer #2 (Part 1)**
>
> We would like to thank the reviewer for the comments.
>
> Q: The authors need to better motivate the advantages of using Shapley value as a data valuation metric. It is not completely clear to me why Shapley value is a good data valuation metric, compared with other options. The authors argue that it is both fair and decomposable (linear in U). However, based on Section 2.2, it is only fair under two extreme cases (identical points and zero marginal contributions). Also, it seems that a lot of other metrics will also satisfy the decomposability condition. Please explain!
>
> A: Thank you for pointing this out. We hope to clarify that the main goal of our paper is to provide theoretical justification for the advantage of the Shapley value over other existing data valuation framework (i.e., leave-one-out error) and show that we can produce an efficient Shapley value calculation heuristic that can benefit multiple applications. While the motivation of using the Shapley value as data value notion has been justified in prior works [1,2,3], inspired by these papers, we leverage the Shapley value as a data value measure in our paper.
>
> In particular, although there are multiple data value measures which may satisfy each single property listed in the paper independently, the Shapley value is the most promising because it is the *only* one that *simultaneously* satisfies the three properties: fairness, decomposability, and group rationality. The first two properties are discussed in the paper, while the last one states that the Shapley values of different data sources add up to the total utility of the machine learning model trained on the collection of these data sources. The third property is interesting in the data market context, because any group of rational data contributors would expect to distribute the full yield of their coalition. We did not extensively discuss the third property in the paper because it is not crucial when we only care about data value in relative terms. We have added the details about the third property into the revised version and provided more motivation for using the Shapley value. Please see Section 2.2.
>
>
> Q: Section 3 and Section 4.1 focus on U defined in equation (3), in which the testing set is a singleton. It seems to be a major limitation of the paper and it is not clear to me whether or not it is easy to generalize the results in these two sections to the general testing set with multiple points. Please explain!
>
> A: Thanks for the question! The result in Section 3 is generalizable to test set with multiple points due to the decomposability property of the Shapley value. That is, the Shapley value of a training point with respect to multiple test instances is the sum of the Shapley values with respect to each test instance. The result in Section 4 can also be generalized to multiple test point setting using the decomposability property. Specifically, for any two training points, the $K$NN Shapley value with respect to multiple validation points is order-preserving when the order remains the same on each validation point, while the $K$NN LOO value with respect to multiple validation points is order-preserving when the two points are within the $K$-nearest neighbors of all validation points and the order remains the same on each validation point. We can see that similar to the single-validation-point setting, the condition for the $K$NN LOO value with respect to multiple validation points to be order-preserving is more stringent than that for the KNN Shapley value.  We have incorporated the discussion of the extension to multiple test points in the Section 3.1 and 4.1.

---

> > ### Author Response · Authors · 2019-11-11
> > **Response to Reviewer #2 (Part 2)**
> >
> > Q: In Definition 3, the definition for the dummy point. This definition requires that U(S \union {z_i}) = U(S) for any S \subseteq D, and in particular it should hold for S=\emptyset. Does U({z_i}) = U(\emptyset) make sense in most practical problems?
> >
> > A: Thank you for pointing it out. Indeed, our result in Theorem 3 does not require the existence of dummy points in the training set. We introduce the concept of dummy points only to better illustrate the implication of Theorem 3. Because both the Shapley value and the LOO value are zero at dummy points, Theorem 3 can be restated as follows:
> > —
> > For a learning algorithm A(·) that achieves (\epsilon(N), \delta(N))-DP when training on N data points. Let the performance measure be U(S) = − 1/M \sum_{i=1}^M E_{h~A(S)} l(h, z_{test,i}) for S \subseteq D. Let  \epsilon’(N) = e^{c(N)} − 1 + ce^{c\epsilon(N)}\delta(N). Then, it holds that
> > \max_{z_i\in D} \nu_{loo}(z_i) \leq \epsilon’(N-1)
> > \max_{z_i\in D} \nu_{shap}(z_i) \leq \frac{1}{N-1} \sum_{i=1}^{N-1} \epsilon’(i)
> > —
> > Essentially, our theorem wants to show that for differentially private learning algorithms, the values of both bad and good points both converge to zero when the training size is large. However, compared with the LOO value, the convergence is slower for the Shapley value; therefore, the Shapley value provides better chance to differentiate good points from the bad. We have revised Section 4.2 to make it clear.
> >
> > We have also polished writing and fixed the typos.
> >
> > [1] https://arxiv.org/abs/1902.10275
> > [2] https://arxiv.org/abs/1908.08619
> > [3] https://arxiv.org/abs/1904.02868

---

### Decision · Program_Chairs · 2019-12-19

**Decision:**

Reject

**Comment:**

There is insufficient support to recommend accepting this paper.  The authors provided detailed responses to the reviewer comments, but the reviewers did not raise their evaluation of the significance and novelty of the contributions as a result.  The feedback provided should help the authors improve their paper.